# Microbial colonization induces histone acetylation critical for inherited gut-germline-neural signaling

**Chunlan Hong‡, Jonathan Lalsiamthara‡, Jie Ren, Yu Sang, Alejandro Aballay***

Department of Molecular Microbiology and Immunology, Oregon Health & Science University, Portland, Oregon, United States of America

‡ These authors share first authorship on this work.
* aballay@ohsu.edu

**Data Availability Statement:** All relevant data are within the paper and its Supporting Information files.

## Abstract

The gut-neural axis plays a critical role in the control of several physiological processes, including the communication of signals from the microbiome to the nervous system, which affects learning, memory, and behavior. However, the pathways involved in gut-neural signaling of gut-governed behaviors remain unclear. We found that the intestinal distension caused by the bacterium *Pseudomonas aeruginosa* induces histone H4 Lys8 acetylation (H4K8ac) in the germline of *Caenorhabditis elegans*, which is required for both a bacterial aversion behavior and its transmission to the next generation. We show that induction of H4K8ac in the germline is essential for bacterial aversion and that a 14-3-3 chaperone protein family member, PAR-5, is required for H4K8ac. Our findings highlight a role for H4K8ac in the germline not only in the intergenerational transmission of pathogen avoidance but also in the transmission of pathogenic cues that travel through the gut-neural axis to control the aversive behavior.

## Introduction

Increasing evidence suggests that the intestine plays an important role in response to environmental changes, which ultimately affect behaviors by communicating with neurons [1–5]. While there is also evidence indicating that microbial cues sensed by the intestine can be transmitted to the offspring and affect their behavior [6,7], the pathways involved in gut-neural communication and the inheritability of gut-governed behaviors remain unclear. The germline can transmit epigenetic information from the environment to the next generation through communication with other tissues [8,9], and it may regulate behaviors in response to environmental stress [10,11]. However, a potential role of the germline in the gut-neural axis has not been established.

To provide insights into the gut-neural circuits that regulate behaviors in response to microbial colonization of the intestine, we have taken advantage of the nematode *Caenorhabditis elegans*, which has evolved behavioral responses that allow the animal to avoid potentially pathogenic bacteria. Upon exposure to *Pseudomonas aeruginosa*, *C. elegans* exhibits a pathogen-aversive behavior, which is governed by distinct groups of neurons [12–15]. Moreover,

**Funding:** This work was supported by the National Institute of Allergy and Infectious Diseases grant number AI117911 (AA) and the National Institute of General Medical Sciences grant number GM070977 (AA). The funders had no role in study design, data collection and analysis, decision to publish, or preparation of the manuscript.

**Competing interests:** The authors have declared that no competing interests exist.

**Abbreviations:** BiFC, bimolecular fluorescence complementation; ChIP-MS, chromatin immunoprecipitation-mass spectrometry; DMP, defecation motor program; GFP, green fluorescent protein; H4K8ac, histone H4 Lys8 acetylation; H3K4me1, monomethylation of histone H3 Lys4; H3K4me3, trimethylation of histone H3 Lys4; PTM, posttranslational modifications; RNAi, RNA interference.

recent studies indicate that *P. aeruginosa* colonization of the intestine causes a distension that regulates behavior and learning via neuroendocrine signaling [16,17].

In this study, we show that the germline is part of the gut-neural axis involved in pathogen avoidance. The mechanism through which intestinal colonization by bacteria induces pathogen avoidance requires histone H4 Lys8 acetylation (H4K8ac) in the germline. H4K8ac is also needed for the transmission of the pathogen-aversive behavior to the next generation. Chromatin immunoprecipitation-mass spectrometry (ChIP-MS) identified a 14-3-3 chaperone protein family member, PAR-5, as essential in the germline for H4K8ac and gut-neural signaling of pathogen avoidance. These results suggest that H4K8ac in the germline participates in a circuit that receives inputs from the infected gut and transmits the information to the nervous system to elicit pathogen avoidance.

## Results and discussion

### Microbial colonization of the intestine induces H4K8ac in the germline

Histone posttranslational modifications (PTM) are the most common epigenetic mechanisms, and different modifications have been found to be involved in diverse biological processes across species, including *C. elegans* [18–20]. Methylation and acetylation are common histone PTM that generally affect gene expression by altering the activity of origins of DNA replication or chromatin structure and gene transcription [21,22]. As a first step to studying whether histone PTM play a role in the control of the pathogen-aversive behavior elicited by microbial colonization of the *C. elegans* intestine, we looked at monomethylation of histone H3 Lys4 (H3K4me1), trimethylation of histone H3 Lys4 (H3K4me3), and histone H4K8ac as they have been linked to immunological memory in plants and mammals [23,24]. First, we studied these histone modifications in young adult animals that were exposed to *P. aeruginosa* for 24 hours. To ensure that the histone PTM analyzed do not correspond to the progeny of the animals, we used *fer-1* animals, which are infertile at 25°C due to a mutation that prevents the sperm from penetrating the oocyte [25]. We found that only H4K8ac increased (Fig 1A and S1A Fig), suggesting that *P. aeruginosa* infection induces it in the infected animals.

Because infection by *P. aeruginosa* correlates with colonization and distension of the *C. elegans* intestinal lumen, which triggers bacterial aversion [16,26], we reasoned that H4K8ac may also increase as a consequence of intestinal distension. To test this hypothesis, we studied H4K8ac in *aex-5* and *eat-2* RNA interference (RNAi) animals. Inhibition of genes *aex-5* and *eat-2*, which alters the defecation motor program (DMP) of the animals, results in pathogen avoidance triggered by intestinal distension [16]. Consistent with the idea that intestinal distension alone induces H4K8ac, inhibition of *aex-5* and *eat-2* resulted in induced H4K8ac in uninfected animals (Fig 1B, S1B and S1C Fig).

To determine where histone acetylation occurs, we performed whole animal fluorescent immunohistochemistry using an antibody that recognizes H4K8ac. We found that H4K8ac increased mainly in the germline upon exposure to *P. aeruginosa* (Fig 1C) or inhibition by RNAi of *aex-5* or *eat-2* (Fig 1D, S1D Fig). These results highlight an important role of the germline in communicating danger signals from the intestine to the nervous system to elicit pathogen avoidance.

### Gut-germline-neural signaling is required for pathogen avoidance

We sought to address whether the germline was part of the gut-neural signaling required for the elicitation of pathogen avoidance. To study a potential role of the germline in gut-neural signaling, we used *glp-1* animals, which lack most germline cells due to defects in mitotic and meiotic division [27,28]. As shown in Fig 2A and S2A Fig, *P. aeruginosa*-induced H4K8ac was

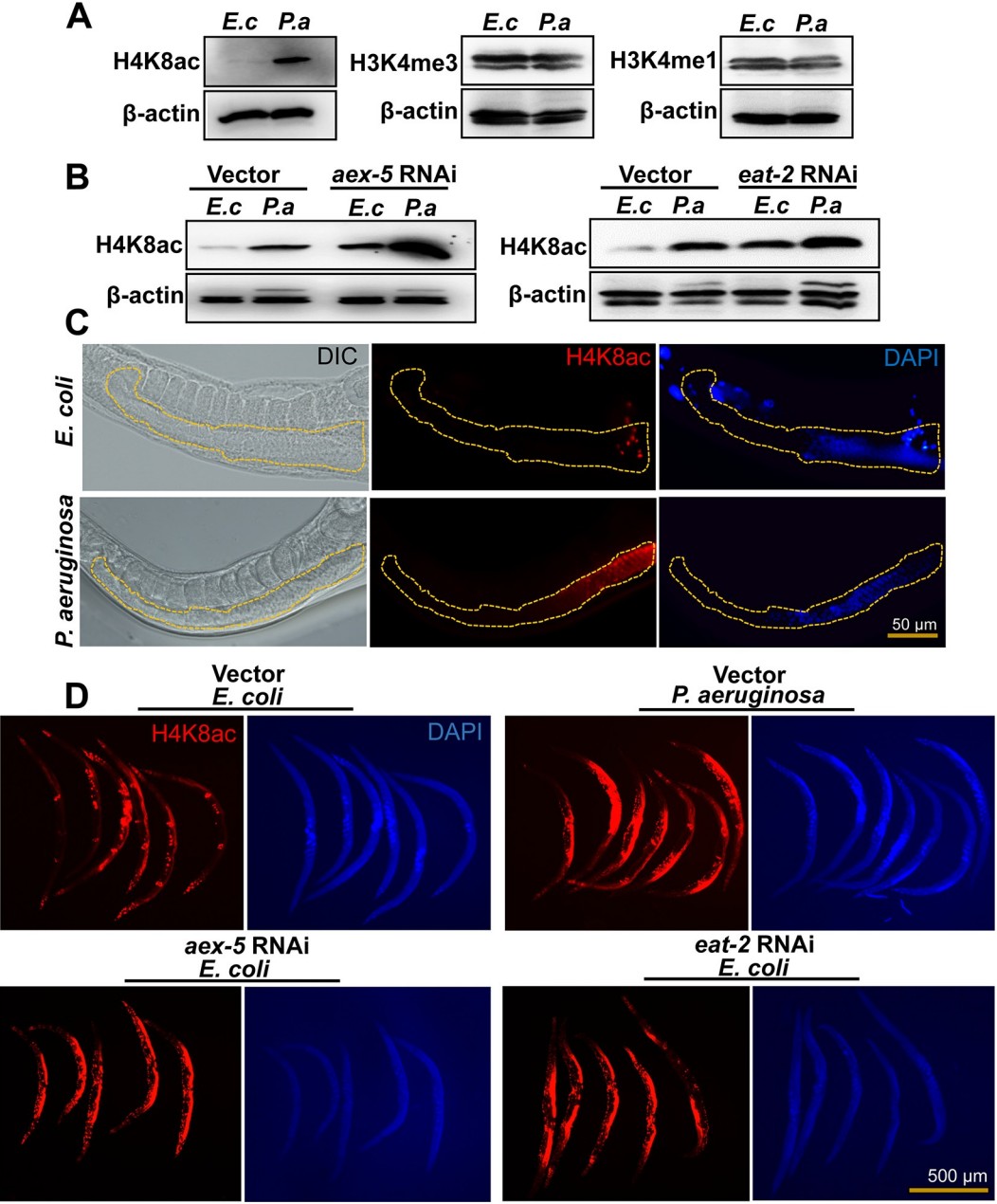

**Fig 1. *P. aeruginosa* infection and intestinal distension induce H4K8ac in the germline.** (A) Western blots of extracts from *fer-1(b232)* animals exposed to *E. coli* (*E. c*) or *P. aeruginosa* (*P. a*) for 24 hours at 25°C ($n \approx 1,000$; representative of 3 independent experiments). (B) Western blots on extracts from *fer-1(b232)* animals exposed to *E. coli* (*E. c*) or *P. aeruginosa* (*P. a*) following *aex-5* and *eat-2* RNAi for 24 hours at 25°C ($n \approx 1,000$; representative of 3 independent experiments). The *fer-1(b232)* animals were maintained at 15°C. To induce sterility, L1-stage animals were transferred to 25°C and allowed to develop. L4-stage animals were then transferred to RNAi plates and allowed to grow for 24 hours at 25°C. "n" represents the number of animals for each experiment (A, B). (C) Representative microscopic images of portions of *C. elegans* germline depicting differences in H4K8ac patterns. Yellow-dotted lines were used to outline the germline. (D) Whole-mount immunofluorescence profile of wild-type animals stained with anti-H4K8ac antibody, post exposure to *E. coli* (*E. c*) or *P. aeruginosa* (*P. a*) following *aex-5* and *eat-2* RNAi for 24 hours at 25°C. See S1 Raw Images for uncropped immunoblot images. H4K8ac, histone H4 Lys8 acetylation; RNAi, RNA interference.

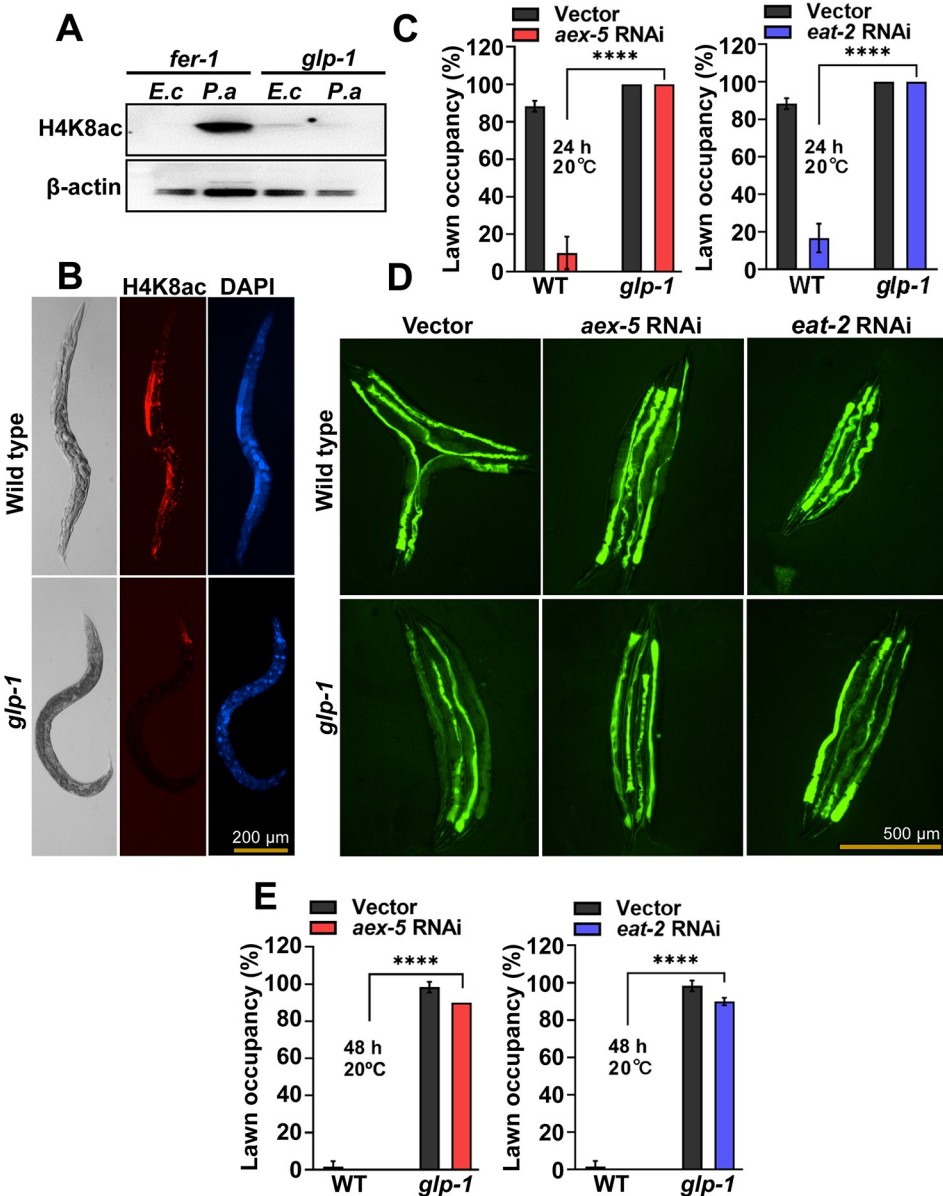

**Fig 2. The germline is part of the gut-neural axis involved in pathogen avoidance.** (A) Western blot of extracts from *fer-1(b232)* and *glp-1(e2141)* animals exposed to *E. coli* (*E. c*) or *P. aeruginosa* (*P. a*) for 24 hours at 25°C ($n \approx 1,000$; representative of 3 independent experiments). (B) Representative microscopic images of wild-type N2 and *glp-1(e2141)* animals stained with anti-H4K8ac antibody following exposure to *P. aeruginosa* for 24 hours at 25°C. (C) Lawn occupancy of wild-type N2 (WT) or *glp-1(e2141)* animals at 24 hours of exposure to *P. aeruginosa* at 20°C following *aex-5* and *eat-2* RNAi ($n = 20$). Three independent experiments were performed. "n" represents the number of animals for each experiment. "*" asterisk indicates significant difference; **** $P \leq 0.0001$. (D) Bacterial colonization of wild-type N2 and *glp-1(e2141)* animals after 48 hours of exposure to *P. aeruginosa* at 20°C following *aex-5* and *eat-2* RNAi ($n = 4$). (E) Lawn occupancy of wild-type N2 (WT) or *glp-1(e2141)* animals at 48 hours of exposure to *P. aeruginosa* at 20°C following *aex-5* and *eat-2* RNAi ($n = 20$). Three independent experiments were performed. "n" represents the number of animals for each experiment. "*" asterisk indicates significant difference; **** $P \leq 0.0001$. The *fer-1(b232)* and *glp-1(e2141)* animals were maintained at 15°C. To induce sterility, L1-stage animals were transferred to 25°C and allowed to develop into young adults and subjected to the corresponding assays. For RNAi induction, L4-stage animals were transferred to RNAi plates and allowed to grow for 24 hours at 25°C. See S1 Raw Images for uncropped immunoblot images and S1 Data for the corresponding data. H4K8ac, histone H4 Lys8 acetylation; RNAi, RNA interference; WT, wild type.

not observed in *glp-1* animals. We also confirmed the lack of H4K8ac by immunohistochemistry (Fig 2B, S2B Fig), which also indicates that histone acetylation indeed occurs in the germline in response to intestinal distension caused by microbial colonization.

Because H4K8ac is induced in the germline of DMP-defective animals, which exhibit rapid pathogen avoidance, and *glp-1* animals lack most germline cells and exhibit no H4K8ac, we hypothesized that intestinal distention in *glp-1* animals would fail to elicit avoidance to *P. aeruginosa*. As shown in Fig 2C, inhibition of *aex-5* and *eat-2* did not elicit pathogen avoidance in *glp-1* animals. It is known that *glp-1* animals exhibit enhanced resistance to a wide array of microbes, including *P. aeruginosa* [29], which results in a slow microbial colonization [30]. Thus, it was not clear whether the inability of *glp-1* animals to avoid *P. aeruginosa* was due to the absence of the germline or the enhanced resistance to infection and colonization by the pathogen. To distinguish between these two possibilities, we tested pathogen avoidance after 48 hours, when bacterial colonization was comparable in wild-type and *glp-1* animals deficient in *aex-5* and *eat-2* (Fig 2D, S2C Fig). We did not observe any difference in the pathogen avoidance of *glp-1* animals compared to that of *glp-1* animals deficient in *aex-5* and *eat-2* (Fig 2E), even at times when they were similarly colonized by *P. aeruginosa* (Fig 2D, S2C Fig).

To further confirm the relationship between bloating-mediated avoidance behavior and H4K8ac, we measured acetylation levels in animals deficient in the *nol-6* gene. Previous studies have shown that the RNAi of *nol-6*, a nucleolar RNA-associated protein-encoding gene, reduces bloating of the intestinal lumen caused by bacterial infection [30]. This reduction in *nol-6* expression results in delayed pathogen avoidance [17]. We found that *nol-6* RNAi suppressed the enhanced H4K8ac in the germline of *P. aeruginosa*-infected animals (S2D and S2E Fig). Taken together, these results indicate that intestinal distension caused by *P. aeruginosa* infection enhances H4K8ac in the germline that is required for pathogen avoidance.

## PAR-5 is required for H4K8ac in the germline

To identify potential interacting partners that may affect H4K8ac in response to *P. aeruginosa* colonization, we performed ChIP-MS. A total of 25 H4K8 acetylated-interacting candidate proteins that were up-regulated more than 3-fold in infected animals were identified (S1 Table). We decided to further study PAR-5 because out of all the H4K8 acetylated-interacting candidate proteins that are up-regulated more than 3-fold by *P. aeruginosa* infection, it is the only one that is expressed in the germline and in neurons, from where it could also be involved in the control of pathogen avoidance. Another reason why we focused on PAR-5 is that it belongs to a 14-3-3 family of chaperones [31,32] that, through interactions with different proteins, can regulate PTM such as H4K8ac.

We confirmed the direct binding of PAR-5 and H4 using coimmunoprecipitation (S3A Fig). We also confirmed the protein–protein interaction in vivo using bimolecular fluorescence complementation (BiFC), which allows for the determination of physical interactions of proteins in living cells through direct visualization [33]. The BiFC constructs were engineered to individually express, under the control of the heat shock promoter P*hsp-16.41*, green fluorescent protein (GFP) fragments translationally fused with PAR-5 and H4, which is a *C. elegans* ortholog of human H4. The interaction between the two proteins would bring the nonfluorescent fragments into close proximity for reconstitution and fluorescence. Twelve hours after heat shock, we observed fluorescence, indicating a physical interaction between PAR-5 and H4 in vivo (Fig 3A). Animals carrying BiFC constructs without H4 did not exhibit fluorescence. Knockdown of *par-5* by RNAi resulted in a significant reduction of fluorescence (Fig 3B and 3C), further confirming that the presence of the two proteins is required for the GFP reconstitution. As shown in Fig 3A and 3D, the protein interaction occurs in the nuclei of

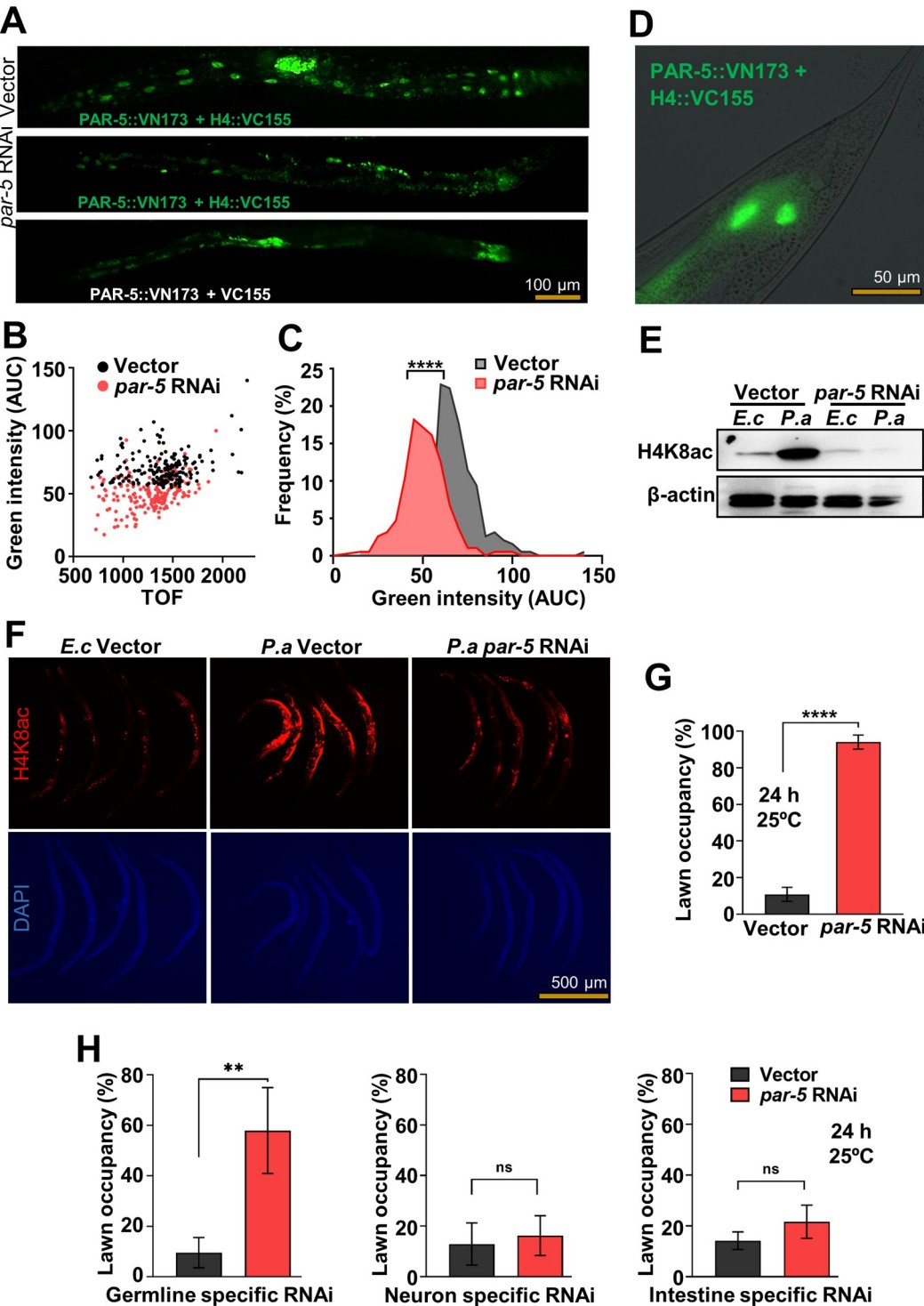

**Fig 3. PAR-5 is required for *P. aeruginosa* H4K8ac in the germline.** (A) Representative microscopic images of vector control or *par-5* RNAi animals expressing BiFC constructs 12 hours after heat shock at 33˚C. Control animals without the *H4*::VC155 construct were used to establish the background fluorescence. (B) Dot-plot representation of green fluorescence intensity versus TOF of vector or *par-5* RNAi BiFC animals. (C) Frequency distribution of green fluorescence-AUC of vector and *par-5* RNAi transgenic BiFC animals. Three independent experiments were performed. "*" indicates significant difference; **** $P \leq 0.0001$. (D) High-magnification fluorescent micrograph of nuclear localization of PAR-5 protein, post 12 hours heat shock recovery. (E) Western blot of extracts from *fer-1(b232)* animals exposed to *E. coli* (*E. c*) or *P. aeruginosa* (*P. a*) for 24 hours at 25˚C following *par-5* RNAi at 25˚C ($n \approx 1,000$; representative of 3 independent experiments). The *fer-*

*1(b232)* and *glp-1(e2141)* animals were maintained at 15˚C. To induce sterility, L1-stage animals were transferred to 25˚C and allowed to develop into L4s. L4 animals were transferred to RNAi plates and allowed to grow for 24 hours at 25˚C. (F) Whole-mount immunofluorescence profile of wild-type animals stained with anti-H4K8ac, post exposure to *E. coli* (*E. c*) or *P. aeruginosa* (*P. a*) for 24 hours at 25˚C following *par-5* RNAi for 24 hours at 25˚C. (G) Lawn occupancy of wild-type N2 animals at 24 hours of exposure to *P. aeruginosa* following *par-5* RNAi at 25˚C (*n* = 20). Three independent experiments were performed. "n" represents the number of animals for each experiment. "*" asterisk indicates significant difference; **** $P \leq 0.0001$. (H) Lawn occupancy of tissue-specific RNAi animals at 24 hours of exposure to *P. aeruginosa* following *par-5* RNAi in the germline, neurons, or the intestine at 25˚C (*n* = 20). Four independent experiments were performed. "n" represents the number of animals for each experiment. "*" asterisk indicates significant difference; "ns" indicates nonsignificant; ** $P \leq 0.005$. See S1 Raw Images for uncropped immunoblot images and S1 Data for the corresponding data. AUC, area under the curve; BiFC, bimolecular fluorescence complementation; H4K8ac, histone H4 Lys8 acetylation; RNAi, RNA interference; TOF, time of flight.

*hsp-16.41*-expressing cells. Even though PAR-5 is required for development and its inhibition may have wide effects on the germline that might indirectly affect H4 acetylation, our results indicate that PAR-5 directly interacts with histone.

Consistent with this idea that PAR-5 regulates H4K8ac, we found that *par-5* RNAi inhibited the induction of H4K8ac caused by *P. aeruginosa* infection (Fig 3E and S3B Fig). We also found that *par-5* RNAi inhibited H4K8ac in the germline (Fig 3F and S3C Fig). To further confirm the relationship between H4K8ac and pathogen avoidance, we asked whether animals fail to avoid *P. aeruginosa* when *par-5* is inhibited. As shown in Fig 3G, animals did not avoid *P. aeruginosa* when *par-5* was inhibited by RNAi. Because *par-5* and the homolog gene *ftt-2* share approximately 78.2% sequence identity at the nucleotide level and approximately 85.9% sequence identity at the amino acid level [34], we studied the specificity of *par-5* RNAi. First, we investigated the pathogen avoidance of *ftt-2* mutants and found that unlike *par-5* RNAi animals, *ftt-2* animals were capable of avoiding *P. aeruginosa* (S4A Fig). We also investigated the expression of the two proteins using anti-FTT-2 and anti-PAR-5 antibodies and found that PAR-5 but not FTT-2 diminished upon *par-5* RNAi (S4B and S4C Fig). Our results indicate that enhanced H4K8ac in the germline is required for pathogen avoidance. Thus, we employed strains capable of tissue-specific RNAi to evaluate the tissue-specific contributions of *par-5* RNAi responsible for the inhibition of pathogen avoidance. As shown in Fig 3H, *par-5* RNAi in the germline significantly reduced pathogen avoidance, which is consistent with our previous results and suggests that H4K8ac occurs in the germline in response to infection. Consistent with this idea, *par-5* RNAi in the germline, but not in the intestine or in neurons, significantly suppressed the *P. aeruginosa*-induced H4K8ac (S5 Fig). Whole animal fluorescent immunohistochemistry confirmed that *P. aeruginosa*-induced H4K8ac is inhibited by *par-5* RNAi in the germline (S6 Fig).

## Enhanced H4K8ac in the germline induces an intergenerational pathogen avoidance behavior

We hypothesized that if the intestinal distension caused by bacterial colonization or inhibition of DMP genes induces H4K8ac in the germline, the signal may be transmitted to the progeny. Thus, we asked whether the offspring of animals exposed to *P. aeruginosa* could also exhibit increased H4K8ac in the germline. Because the effect of RNAi is transgenerationally transmitted, we cannot investigate whether H4K8ac induced by inhibition of DMP genes is also passed to the progeny. Therefore, we used heat-killed *E. coli* that induces intestinal distension, similar to that of DMP-defective animals, and also elicits a similar pathogen avoidance behavior [16]. We exposed L4 animals to *P. aeruginosa* or heat-killed *E. coli* for 24 hours, two conditions which can induce bloating in the intestinal lumen and result in bacterial aversion [16]. As shown in Fig 4A and S7 Fig, the F1 offspring from infected P0 animals exhibited higher

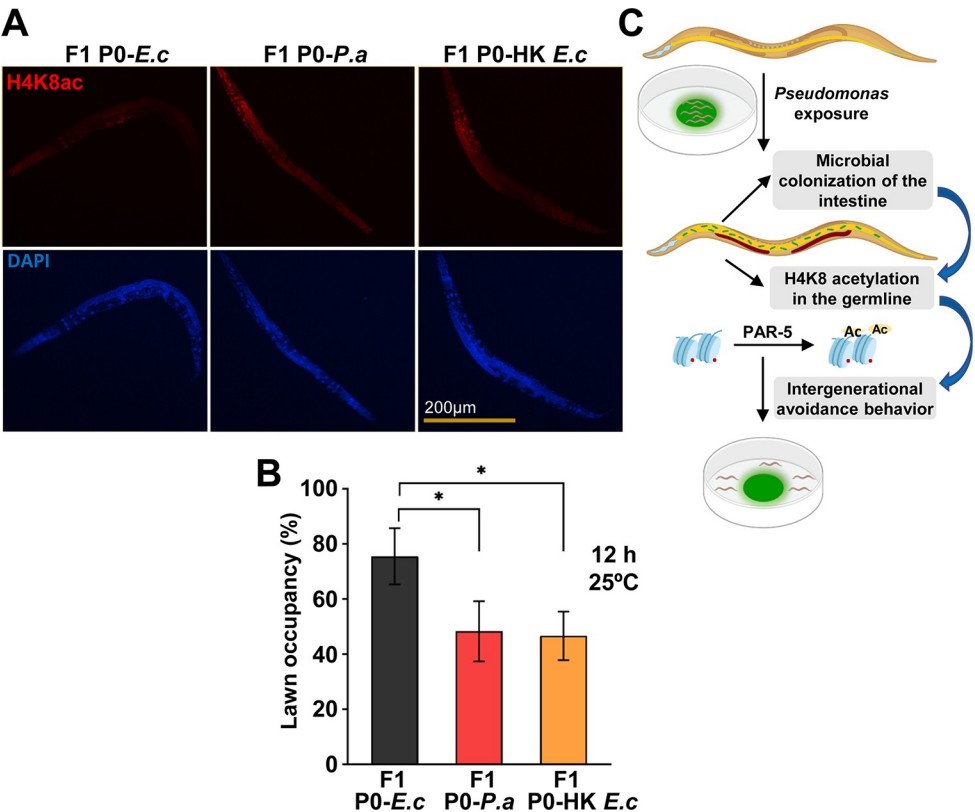

**Fig 4. Enhanced H4K8ac in the germline is required for the intergenerational pathogen avoidance.** (A) Whole-mount immunofluorescence using an anti-H4K8ac antibody to strain F1 wild-type N2 young adults from P0 animals exposed to *E. coli* (*E. c*), *P. aeruginosa* (*P. a*), or heat-killed *E. coli* (HK *E. c*) starting at the L4 stage for 24 hours. (B) Lawn occupancy of F1 wild-type N2 animals at 12 hours from P0 animals exposed to *E. coli* (*E. c*), *P. aeruginosa* (*P. a*), or heat-killed *E. coli* (HK *E. c*) starting at the L4 stage for 24 hours ($n$ = 20). Three independent experiments were performed. "n" represents the number of animals for each experiment. "*" asterisk indicates significant difference; $^{*}P \leq 0.05$. (C) *C. elegans* exposure to *P. aeruginosa* causes an intestinal colonization and distention that leads to H4K8ac in the germline. Both H4K8ac in the germline and the germline itself are required for an intergenerational pathogen avoidance. See S1 Data for the corresponding data. H4K8ac, histone H4 Lys8 acetylation.

H4K8ac in the germline than control animals, indicating that bloating of the intestine induces H4K8ac in the germline that is intergenerationally transmitted.

Intestinal distension caused by bacterial colonization or inhibition of DMP genes induces pathogen avoidance [16,17] and H4K8ac in the germline (Fig 1D). Moreover, the enhanced H4K8ac is transmitted to the F1 offspring (Fig 4A and S7 Fig). Thus, we investigated whether the F1 offspring also exhibits higher pathogen avoidance. The progeny of animals infected with *P. aeruginosa* or fed heat-killed *E. coli* to induce intestinal distension exhibited significantly higher pathogen avoidance than the progeny of animals fed control live *E. coli*, which does not cause intestinal distension (Fig 4B). Taken together, these results demonstrate that enhanced H4K8ac in the germline is required for the intergenerational pathogen avoidance induced by bloating caused by bacterial colonization of the intestine.

## Conclusions

The gut-neural axis plays a critical role in transmitting signals from the microbiome to the nervous system to respond to environmental changes. We have shown that histone H4K8ac increased in the germline upon exposure to *P. aeruginosa*, suggesting that the intestinal

distension caused by microbial colonization induces histone H4K8ac in the germline. Indeed, we observed that H4K8ac increases in animals that exhibit distended intestines due to *aex-5* or *eat-2* inhibition. The rapid pathogen avoidance elicited by *aex-5* or *eat-2* inhibition is suppressed by the absence of the germline. Furthermore, we found that PAR-5 regulates pathogen avoidance induced by intestinal distension by stabilizing H4K8ac and that reduction of H4K8ac in the germline by *par-5* inhibition suppressed pathogen avoidance (Fig 4C). The inheritance of avoidance elicited by small RNAs from *P. aeruginosa* requires the germline [35,36]. We do not know whether H4K8ac plays a role in the avoidance mediated by small RNAs, which accounts for a fraction of the avoidance elicited by *P. aeruginosa*. Our results highlight a critical role for H4K8ac in the germline in the control of the gut-neural axis in response to *P. aeruginosa* infection. Further studies will be required to identify the downstream signals involved in germline-neural communication.

## Materials and methods

### Bacterial strains

The following bacterial strains were used: *Escherichia coli* OP50, *Pseudomonas aeruginosa* PA14, and *P. aeruginosa* PA14-GFP. Bacterial cultures were grown in Luria-Bertani (LB) broth at 37˚C.

### Nematode strains and growth conditions

*C. elegans* hermaphrodites were maintained on *E. coli* OP50 at 20˚C, except HH142 *fer-1 (b232)*, CB4037 *glp-1(e2141)* strains that were maintained at 15˚C. Bristol N2 was used as the wild-type control. Germline-specific RNAi strain DCL569 (*mkcSi13 II;rde-1(mkc36) V*), neuron-specific RNAi strain TU3401 (*sid-1(pk3321) V;uIs69 V*), gut-specific RNAi strain MGH171 (*sid-1(qt9) V;alxIs9*), HH142 *fer-1(b232)*, CB4037 *glp-1(e2141)*, and MT14355 *ftt-2 (n4426)* strains were obtained from the *Caenorhabditis elegans* Genetics Center (University of Minnesota, Minneapolis).

### Bacterial lawn avoidance assay

The bacterial lawns were prepared by picking individual *P. aeruginosa* PA14 colonies into 3 mL of the LB and growing them at 37˚C for 12 hours on a shaker. Then, a 20-μL culture was seeded onto the center of a 3.5-cm modified NGM plate and incubated at 37˚C for 12 hours. Twenty synchronized hermaphroditic animals grown on *E. coli* HT115(DE3) carrying a control vector or an RNAi clone targeting a gene were transferred into the bacterial lawns, and the number of animals on and off the lawn were counted at the indicated times for each experiment. Experiments were performed at 25˚C except for *aex-5* and *eat-2* RNAi animals, which are hypersusceptible to *P. aeruginosa* at 25˚C [16]. The percent occupancy was calculated as $(N_{on} lawn/N_{total}) \times 100$.

### *P. aeruginosa*-GFP colonization assay

Bacterial lawns were prepared by inoculating individual bacterial colonies into 3 mL of LB with 50 μg/mL kanamycin and growing them at 37˚C for 12 hours on a shaker. For the colonization assays, bacterial lawns of *P. aeruginosa*-GFP were prepared by spreading 200 μL of the culture on the entire surface of 3.5 cm diameter-modified NGM plates. The plates were incubated at 37˚C for 12 hours and then cooled to room temperature before the animals were transferred. Synchronized L1-stage wild-type N2 and *glp-1(e2141)* were transferred to 25˚C. L4-stage animals were then transferred to RNAi plates and allowed to grow for 24 hours at

25˚C. Exposure to *P. aeruginosa*-GFP was conducted at 20˚C for 48 hours. The animals were transferred from *P. aeruginosa*-GFP plates to fresh *E. coli* plates for 10 minutes to eliminate *P. aeruginosa*-GFP adhered to their body. This procedure was repeated 3 times. Subsequently, 10 animals/condition were transferred into 50 μL of PBS containing 0.01% Triton X-100 and grounded with pestle and glass beads. Ten-fold serial dilutions of the lysates ($10^{-1}$, $10^{-2}$, $10^{-3}$, $10^{-4}$) were made and seeded onto LB plates containing 50 μg/mL of kanamycin to select for *P. aeruginosa*-GFP cells and grown overnight at 37˚C. Single colonies were counted the next day, the dilution factors were incorporated into the colony-counts, and the results were represented as colony-forming units (CFU) per animal. Three independent experiments were performed.

## RNA interference

The preparation of RNAi experiments has been explained in previous studies [37]. Briefly, *E. coli*, with the appropriate vectors, was grown in LB broth containing ampicillin (100 μg/mL) at 37˚C overnight and plated onto NGM plates containing 100 μg/mL ampicillin and 3 mM iso-propyl β-d-1-thiogalactopyranoside (IPTG) (RNAi plates). RNAi-expressing bacteria were grown overnight at 37˚C. Synchronized L4-stage animals were transferred to RNAi plates and grown for 24 hours at 25˚C, unless otherwise indicated, before the subsequent experiments. All RNAi clones except *eat-2* were from the Ahringer RNAi library.

## Chromatin immunoprecipitation-mass spectrometry

Total protein extracts from *fer-1(b232)* animals were obtained by sonication in FA buffer after crosslinking with formaldehyde for 30 minutes. Anti-H4K8ac antibody was used to precipitate proteins bound to H4K8ac by incubation overnight followed with magnetic beads incubation at 4˚C for 1 hour to pull down the proteins. Proteins were eluted with sample buffer and resolved on a 4% to 12% NuPage Novex gel (Invitrogen, Waltham, Massachusetts, USA) and stained with Imperial Protein Stain (Thermo Fisher Scientific, Waltham, Massachusetts, USA). Gel was run for 10 minutes. Cut bands were reduced, alkylated with iodoacetamide, and in-gel digested with trypsin (Promega, Madison, Wisconsin, USA) prior to MS analysis.

## Western blot assay

Synchronized *fer-1(b232)* or *glp-1(e2141)* animals were transferred to plates with *P. aeruginosa* for 24 hours with *E. coli* as control. Worms were collected and washed 3 times with M9 to remove the bacteria. The cell lysates were obtained by sonication. The samples were mixed with sample loading buffer for gel electrophoresis. Proteins were transferred from the gel to the membrane at 300 mA for 50 minutes. After 1 hour blocking at room temperature, the membrane was incubated at 4˚C overnight with anti-H4K8ac (ab15823, Abcam, Cambridge, Massachusetts, USA), anti-FTT-2, anti-β-actin (ab8227, Abcam, Cambridge, Massachusetts, USA), and anti-PAR-5 antibodies followed with 1-hour anti-Rabbit antibody at room temperature. β-actin serves as internal control. Anti-FTT-2 and anti-PAR-5 antibodies were gifts from Dr. Andrew Golden. Chemiluminescence signal was detected using ImageQuant LAS 4000 (GE Healthcare, Chicago, Illinois, USA). The densities of the protein bands were quantified using Image J and represented as fold change. Fold change is the ratio of mean density of test sample over control sample after normalization with β-actin.

## Whole mount fluorescent immunohistochemistry

Bristol N2 wild-type animals were used for whole mount fluorescent immunohistochemistry, unless otherwise indicated. Synchronized young adult animals were exposed to *P. aeruginosa*

or *E. coli* for 24 hours at 25°C. Worms were washed 3 times with M9 to remove the bacteria and resuspended in fixing solution (160 mM KCl, 100 mM Tris-HCl (pH 7.4), 40 mM NaCl, 20 mM Na$_2$EGTA, 1 mM EDTA, 10 mM spermidine HCl, 30 mM PIPES (pH 7.4), 1% Triton X-100, 50% methanol, 2% formaldehyde) and subjected to snap freezing in liquid nitrogen. The worms were fixed on ice for 4 hours and washed briefly in T buffer (100 mM Tris-HCl (pH 7.4), 1 mM EDTA, 1% Triton X-100) before a 15-minute incubation in T buffer supplemented with 1% β-mercaptoethanol at 37°C. The worms were washed with borate buffer (25 mM H$_3$BO$_3$, 12.5 mM NaOH (pH 9.5)) and then incubated in borate buffer containing 10 mM DTT for 15 minutes, followed by H$_2$O$_2$ incubation for another 30 minutes. Worms were blocked in PBST (PBS (pH 7.4), 0.5% Triton X-100, 1 mM EDTA) containing 1% BSA for 30 minutes and incubated overnight with anti-H4K8ac antibody (1:100; ab15823, Abcam) and with Alexa Fluor 594 secondary antibody (1:300; ab150080, Abcam). 4′,6-diamidino-2-phenylindole (DAPI; 20 μg/mL) was added to visualize nuclei. The worms were mounted on a microscope slide and visualized using stereofluorescence microscope (Leica M165 FC or DM4-B, Leica, Wetzlar, Germany). The fluorescence intensity was quantified using Image J. The whole animal fluorescence was calculated using the following equation: corrected whole animal fluorescence = integrated density − (area of selected animal × mean fluorescence of background readings).

## Bimolecular fluorescence complementation (BiFC) and plasmid construction

To construct plasmids for the BiFC assay for protein interaction, *par-5* and *his-1* cDNA were subcloned into pCE-BiFC-VN173 and pCE-BiFC-VC155 plasmids (Addgene, Watertown, Massachusetts, USA), which contain the heat shock promoter P*hsp-16.41*. Full-length *par-5* cDNA was subcloned in-frame into pCE-BiFC-VN173 between *BmtI* and *KpnI*, and the full length of the H4-encoding gene *his-1* was subcloned in-frame into pCE-BiFC-VC155 between *BmtI* and *KpnI*. The *his-1* gene encodes for an ortholog of human histone H4, which shares a similar epitope-target that is specific to the anti-H4K8ac antibody (ab15823, Abcam). The BiFC plasmid constructs were injected into N2 worms at 15 ng/μL each, together with coel::RFP at 100 ng/μL (coinjection marker) [33]. To detect the interaction, transgenic animals carrying the BiFC plasmid constructs were raised to young adults at 20°C, heat shocked for 3 hours at 33°C, and allowed to recover for 12 hours at 20°C. Direct visualization of fluorescent signals of the induced expression of fusion proteins (PAR-5 and H4) were captured using a Leica M165 FC fluorescence stereomicroscope. The BiFC assay involving RNAi of *par-5* gene transcript was performed and compared to vector control. Coelomocytes RFP-labelled transgenic animals were first gated using the red channel, and the green fluorescence intensity of transgenic animals was measured using the Copas Biosort instrument (Union Biometrica, Holliston, Massachusetts, USA).

## Statistical analysis

Two-tailed Student *t* test for independent samples was used to analyze the data. For comparing the means of more than two groups, one-way ANOVA with post hoc analysis was performed. All the experiments were repeated at least 3 times and error bars represent the standard deviation, unless otherwise indicated. The data were judged to be statistically significant when $P < 0.05$. "n" represents the number of animals for each experiment. "ns" indicates nonsignificant "*" asterisk indicates significant difference; *$P \le 0.05$; **$P \le 0.005$, ***$P \le 0.0005$, ****$P \le 0.0001$.

## Supporting information

**S1 Fig. *P. aeruginosa* infection and intestinal distension induce H4K8ac in the germline.**
(A) Quantification of band density of western blots assay performed on *fer-1(b232)* animals
exposed to *E. coli* (*E. c*) or *P. aeruginosa* (*P. a*) for 24 hours at 25˚C ($n \approx 1,000$). (B) Quantification of band density of western blots assay performed on *fer-1(b232)* animals exposed to *E. coli*
(*E. c*) or *P. aeruginosa* (*P. a*) for 24 hours at 25˚C following *aex-5* ($n \approx 1,000$) and (C) *eat-2*
RNAi ($n \approx 1,000$). Chemiluminescence signals from samples were detected, and the densities of
the protein bands were quantified and represented as fold change. Fold change is the ratio of
mean density of a given sample over the control *E. coli* sample or the control *E. coli* vector sample for the RNAi assays. The *fer-1(b232)* animals were maintained at 15˚C. To induce sterility,
L1-stage animals were transferred to 25˚C and allowed to develop. L4-stage animals were then
transferred to RNAi plates and allowed to grow for 24 hours at 25˚C. Pathogen exposure was
performed at 25˚C for 24 hours. (D) Quantification of immunofluorescence of wild-type N2
animals stained with anti-H4K8ac antibody post exposure to *E. coli* (*E. c*) or *P. aeruginosa* (*P. a*)
for 24 hours at 25˚C following *aex-5* and *eat-2* RNAi ($n = 5$). Three independent experiments
were performed for the above experiments (A–D, except for H3K4me3 immunoblot assay). "n"
represents the number of animals for each experiment. "*" asterisk indicates significant difference; $^*P \leq 0.05$, $^{**}P \leq 0.005$, $^{***}P \leq 0.0005$, $^{****}P \leq 0.0001$. See S1 Data for the corresponding
data. H4K8ac, histone H4 Lys8 acetylation; H3K4me1, monomethylation of histone H3 Lys4;
H3K4me3, trimethylation of histone H3 Lys4; RNAi, RNA interference.
(TIF)

**S2 Fig. Intact germline is required for pathogen or bloating induced H4K8ac.** (A) Quantification of band density of western blots assay from *fer-1(b232)* and *glp-1(e2141)* animals exposed to
*E. coli* or *P. aeruginosa* for 24 hours at 25˚C ($n \approx 1,000$). Four independent experiments were performed. The densities of the protein bands were quantified and represented as fold change. Fold
change is the ratio of mean density of a given sample over the control *fer-1 E. coli* sample. (B)
Quantification of immunofluorescence of wild-type N2 and *glp-1(e2141)* animals stained with
anti-H4K8ac antibody after exposure to *P. aeruginosa* (*P. a*) for 24 hours at 25˚C ($n = 5$). (C) CFU
of wild-type N2 or *glp-1(e2141)* animals grown on vector control, *aex-5* RNAi, or *eat-2* RNAi
were exposed to *P. aeruginosa*-GFP for 48 hours at 20˚C. Bars represent mean log10 CFU ± SEM.
The *fer-1(b232)* and *glp-1(e2141)* animals were maintained at 15˚C. To induce sterility, L1 animals
were transferred to 25˚C and allowed to develop. L4 animals were then transferred to RNAi plates
and allowed to grow for 24 hours at 25˚C. Pathogen exposure was performed at 25˚C for 24
hours, unless otherwise indicated. (D) Representative microscopic images of wild-type N2 animals
treated with vector control or *nol-6* RNAi and stained with anti-H4K8ac antibody following exposure to *E. coli (E. c)* or *P. aeruginosa* (*P. a*) for 24 hours at 25˚C. (E) Quantification of immunofluorescence of wild-type N2 animals stained with anti-H4K8ac antibody post exposure to *E. coli* (*E.
c*) or *P. aeruginosa* (*P. a*) for 24 hours at 25˚C following *nol-6* RNAi ($n = 5$). Three independent
experiments were performed for the above experiments (B–E). "n" represents the number of animals for each experiment. "ns" indicates nonsignificant; "*" asterisk indicates significant difference; $^*P \leq 0.05$, $^{**}P \leq 0.005$, $^{***}P \leq 0.0005$, $^{****}P \leq 0.0001$. See S1 Data for the corresponding
data. CFU, colony-forming unit; GFP, green fluorescent protein; H4K8ac, histone H4 Lys8 acetylation; RNAi, RNA interference; WT, wild-type.
(TIF)

**S3 Fig. PAR-5 is required for H4K8ac.** (A) Coimmunoprecipitation of PAR-5 using anti-
H4K8ac antibody followed by western blot detection of PAR-5 on *fer-1(b232)* animals exposed
to *E. coli* (*E. c*) or *P. aeruginosa* (*P. a*) for 24 hours at 25˚C following *par-5* RNAi ($n \approx 2,000$).

Error bar represents ±SEM. Three independent experiments were performed. (B) Quantification of band density of western blots assay performed on *fer-1(b232)* animals exposed to *E. coli* or *P. aeruginosa* for 24 hours at 25˚C following *par-5* RNAi ($n \approx 1,000$). Three independent experiments were performed. Chemiluminescence signals from samples were detected; the densities of the protein bands were quantified and represented as fold change. Fold change is the ratio of mean density of a given sample over the control *E. coli* vector sample. (C) Quantification of the immunofluorescence of wild-type N2 animals stained with anti-H4K8ac antibody after exposure to *E. coli* (*E. c*) or *P. aeruginosa* (*P. a*) following *par-5* RNAi ($n = 5$). Three independent experiments were performed. "n" represents the number of animals for each experiment. "*" asterisk indicates significant difference; $^{*}P \leq 0.05$, $^{***}P \leq 0.0005$, $^{****}P \leq 0.0001$. See S1 Raw Images for uncropped immunoblot images and S1 Data for the corresponding data. H4K8ac, histone H4 Lys8 acetylation; RNAi, RNA interference.
(TIF)

**S4 Fig. PAR-5 is required for pathogen avoidance.** (A) Lawn occupancy of wild-type N2 or *ftt-2*(*n4426*) animals at 24 hours following *par-5* RNAi at 25˚C ($n = 20$). Three independent experiments were performed. (B) Western blot detection of FTT-2 or PAR-5 and (C) its quantification on extracts of *fer-1(b232)* animals, exposed to *E. coli* (*E.c*) or *P. aeruginosa* (*P.a*) for 24 hours at 25˚C following *par-5* RNAi ($n \approx 1,000$). Two independent experiments were performed. Chemiluminescence signals from samples were detected, and the densities of the protein bands were quantified and represented as fold change. Fold change is the ratio of mean density of a given sample over the control *E. coli* vector sample. The *fer-1(b232)* animals were maintained at 15˚C. To induce sterility, L1-stage animals were transferred to 25˚C and allowed to develop. L4-stage animals were then transferred to RNAi plates and allowed to grow for 24 hours at 25˚C. "n" represents the number of animals for each experiment. "*" asterisk indicates significant difference; $^{****}P \leq 0.0001$. See S1 Raw Images for uncropped immunoblot images and S1 Data for the corresponding data. RNAi, RNA interference; WT, wild-type.
(TIF)

**S5 Fig. PAR-5 is required for *P. aeruginosa*-mediated H4K8ac in the germline.** Western blot detection and quantification of H4K8ac on different tissue-specific RNAi animals upon *par-5* RNAi and subsequent exposure to *E. coli* (*E. c*) or *P. aeruginosa* (*P. a*) for 24 hours at 25˚C ($n \approx 1,000$). Three independent experiments were performed. Chemiluminescence signals from samples were detected, and the densities of the protein bands were quantified and represented as fold change. Fold change is the ratio of mean density of a given sample over the control *E. coli* vector sample. "n" represents the number of animals for each experiment. "ns" indicates nonsignificant; "*" asterisk indicates significant difference; $^{**}P \leq 0.005$. See S1 Raw Images for uncropped immunoblot images and S1 Data for the corresponding data. H4K8ac, histone H4 Lys8 acetylation; RNAi, RNA interference.
(TIF)

**S6 Fig. PAR-5 is required for *P. aeruginosa*-induced H4K8ac in the germline.** (A) Whole-mount immunofluorescence profiles of tissue-specific RNAi animals stained with anti-H4K8ac antibody. (B) Quantification of immunofluorescence of tissue-specific RNAi animals exposed to *E. coli* (*E. c*) or *P. aeruginosa* (*P. a*) for 24 hours at 25˚C, following *par-5* RNAi induction for 24 hours ($n = 5$). Three independent experiments were performed. "n" represents the number of animals for each experiment. "ns" indicates nonsignificant; "*" asterisk indicates significant difference; $^{*}P \leq 0.05$, $^{**}P \leq 0.005$, $^{***}P \leq 0.0005$. See S1 Data for the corresponding data. H4K8ac, histone H4 Lys8 acetylation; RNAi, RNA interference.
(TIF)

**S7 Fig. Pathogen exposure and intestinal distension of P0 maternal animals increases H4K8ac levels in the germline of F1 offspring.** Quantification of immunofluorescence of H4K8ac levels on F1 progeny from P0 maternal animals exposed to *E. coli* (*E. c*), *P. aeruginosa* (*P. a*), or heat-killed *E. coli* (HK *E. c*) ($n = 20$). Three independent experiments were performed. "n" represents the number of animals for each experiment. "ns" indicates nonsignificant; "*" asterisk indicates significant difference; $^{**}P \leq 0.005$, $^{***}P \leq 0.0005$. See S1 Data for the corresponding data. H4K8ac, histone H4 Lys8 acetylation.
(TIF)

**S1 Raw Images. Original uncropped blot images.**
(PDF)

**S1 Data. Raw data and quantitative observations for all main and supporting figures.**
(XLSX)

**S1 Table. List of histone H4 Lys8 acetylation interacting proteins.**
(PDF)

## Acknowledgments

We thank the Caenorhabditis Genetics Center (Univ. of Minnesota) for the strains used in this study, Dr. Andy Golden (NIDDK, National Institutes of Health, Bethesda, Maryland) for providing anti-FTT-2 and anti-PAR-5 antibodies, Dr. Casey Hoffman and Dr. Abiola O. Olaitan for technical advice, and Dr. Jogender Singh for the suggestions.

## Author Contributions

**Conceptualization:** Chunlan Hong, Jonathan Lalsiamthara, Alejandro Aballay.

**Formal analysis:** Chunlan Hong, Jonathan Lalsiamthara.

**Funding acquisition:** Alejandro Aballay.

**Investigation:** Chunlan Hong, Jonathan Lalsiamthara, Alejandro Aballay.

**Methodology:** Chunlan Hong, Jonathan Lalsiamthara, Alejandro Aballay.

**Project administration:** Alejandro Aballay.

**Resources:** Jie Ren, Yu Sang.

**Supervision:** Alejandro Aballay.

**Validation:** Jonathan Lalsiamthara, Jie Ren, Yu Sang.

**Visualization:** Chunlan Hong, Jonathan Lalsiamthara.

**Writing – original draft:** Chunlan Hong, Alejandro Aballay.

**Writing – review & editing:** Chunlan Hong, Jonathan Lalsiamthara, Jie Ren, Yu Sang, Alejandro Aballay.

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
