## [Editor Report · Decision Letter 0]

10 Jul 2020

Dear Dr Aballay, 

Thank you for submitting your manuscript entitled "Microbial colonization induces histone acetylation critical for inherited gut-germline-neural signaling" for consideration as a Short Reports by PLOS Biology.

Your manuscript has now been evaluated by the PLOS Biology editorial staff as well as by an academic editor with relevant expertise and I am writing to let you know that we would like to send your submission out for external peer review.

Please re-submit your manuscript within two working days, i.e. by Jul 13 2020 11:59PM.

Kind regards,

Ines

--

Ines Alvarez-Garcia, PhD

Senior Editor

PLOS Biology

Carlyle House, Carlyle Road

Cambridge, CB4 3DN

+44 1223–446970

---

## [Decision Letter · Decision Letter 1]

14 Sep 2020

Dear Dr Aballay,

Thank you very much for submitting your manuscript "Microbial colonization induces histone acetylation critical for inherited gut-germline-neural signaling" for consideration as a Short Report at PLOS Biology. Thank you also for your patience as we completed our editorial process, and please accept my sincere apologies for the delay in providing you with our decision. Your manuscript has been evaluated by the PLOS Biology editors, an Academic Editor with relevant expertise, and by three independent reviewers.

As you will see, the reviewers find the conclusions interesting and significant for the field, however they also raise several concerns that should be addressed before we can consider the manuscript for publication. While both Reviewers 2 and 3 mainly ask for several controls, the use of appropriate statistical tests and clarifications, Reviewer 1 finds the conclusions correlative rather than causative and suggests several experiments to address this issue. After discussing the reviews with the academic editor, we think the concerns raised by this reviewer are valid. Although we note this reviewer missed the details included in the manuscript about how the ChIP-MS was performed, the concerns raised need to be addressed experimentally. Specifically, you would need to add data to strengthen the connection between pathogen avoidance and the germline, and to more convincingly demonstrate that PAR-5 interacts with H4K8ac. Depending on how some of the major comments of this reviewer are addressed, we would consider the merit of converting the manuscript into a Research Article.

In light of the reviews (attached below), we will not be able to accept the current version of the manuscript, but we would welcome re-submission of a revised version that takes into account the reviewers' comments. We cannot make any decision about publication until we have seen the revised manuscript and your response to the reviewers' comments. Your revised manuscript is also likely to be sent for further evaluation by the reviewers.

We expect to receive your revised manuscript within 3 months. 

**IMPORTANT - SUBMITTING YOUR REVISION**

*Re-submission Checklist*

*Published Peer Review*

*PLOS Data Policy*

*Blot and Gel Data Policy*

Sincerely,

Ines

--

Ines Alvarez-Garcia, PhD,

Senior Editor,

ialvarez-garcia@plos.org,

PLOS Biology

Reviewers' comments

Rev. 1:

Hong and Aballay present a study that describes H4K8ac as a epigenetic mark in response to exposure to the pathogen PA14. They've shown that there is an increase in H4K8ac marks following pathogen exposure. The authors use a combination of genetics, RNAi, and biochemistry to define a germline response to PA14 with the ability to impact subsequent generations. In addition, the authors define additional players in system that to begin developing a pathway for this innate immunity response. While the data is presented clearly, there are some concerns with regard to interpretation of the results and the methodologies used that reduce enthusiasm for this short article. This potential importance of this finding and the need to identify additional mechanistic details suggest this should be a full length article as in its present form many of the conclusion are correlative rather than causative.

Major concerns:

The biggest drawback to this study is the lack of identification of the enzymes responsible for the addition of H4K8ac marks.

Using glp-1 completely gets rid of germline so there could be confounding factors that affect pathogen avoidance, which are known. Why not use germline restricted RNAi or other germcell manipulations (or gonad loss mutants) to strengthen this section? Although glp-1 mutants can indeed reduce the number of germ cells, they do not abolish all germ cells (it is incomplete). Intriguingly, glp-1 mutants appear to have a low-level of H4K8ac (Figure 2A) with E.coli that does not increase with PA14?

The identification of PAR-5 is not clear. It was identified by ChIP-ms? More details are needed here as I assume this was done with the H4K8ac antibody, but it isn't clear why this would directly pull down PAR-5. The authors should show that PAR-5 is binding directly to H4K8ac or define what it is interacting with. What is missing is a clear identification of what is PAR-5 regulating since it is a cytoplasmic chaperone. Since PAR-5 is important for embryo development and knockdown of par-5 could affect the germline in other ways as well the connection to H4K8ac is confounded.

Based on the IF results, there appears to be some somatic cells with H4K8ac marks? Could the authors look at the levels of H4K8ac marks either through WB or IF for the tissue specific RNAi.

The connection to pathogen avoidance is the weakest as the germline has been shown to impact this response. The authors could look at H4K8ac levels in ftt-2 knockdown worms too, since they share so much homology with PAR-5 but don't affect avoidance.

Lastly, the authors should check what happens to H4K8ac marks following PA14 exposure if intestinal distension is suppressed like with nol-6 RNAi. If H4K8ac marks don't appear if intestinal distension is suppressed that would help back their claim.

Minor concerns:

There are several temperature sensitive mutants used, but it is sometimes unclear what temperatures were used for raising animals in development, versus adulthood, verus experiment. Most seem to be done at 25 (with some exceptions), but the methods state that experiments were done at 20, unless otherwise indicated. It would be nice to have this explicitly outlines for each experiments.

The authors include a whole section where they introduce PAR-5 and how 14-3-3 chaperones are involved with PTMs such as H4K8ac without citations.

Similarly, the transgenerational marks for H4K8ac are cool, but they should reference work that has already shown transgenerational inheritance of pathogen avoidance and specifically Coleen Murphy's work as possible future targets of study or H4K8ac inheritance.

Rev. 2:

Although many bacterial infections are restricted to the intestine, there is increasing evidence that infection causes signaling between different tissues. C. elegans is a powerful system to study bacterial infections, pathogen avoidance, and effects that are passed down between generations. In this manuscript the authors investigate the connection between the intestine and germline, and how this effects animal behavior. The authors find that infection with Pseudomonas aeruginosa induces H4Kac methylation in the germline. This methylation also occurs by knocking down several genes that cause intestinal distension. These RNAi conditions also induce pathogen avoidance which is dependant on the germline. The authors also identify PAR-5 as interacting with H4Kac. The authors show that PAR-5 is necessary for H4Kac methylation in the germline and for bacterial avoidance. Finally, the authors show that under conditions that induce this methylation, that it can be passed on to the next generation and the progeny have increased bacterial avoidance. This is a very interesting and well-carried out study that provides new insight into the connection between the intestine and germline and how communication between these tissues can influence behavior.

Major point:

1. The authors claim that "results demonstrate that enhanced H4K8ac in the germline is required for the transgenerational pathogen avoidance induced by bloating caused by bacterial colonization of the intestine" and claim in 4c that PAR-5 is required for this transgenerational effect. The authors do not show that animals that lack H4K8ac in the germline generate progeny that are defective for bacterial avoidance. They also don't show that PAR-5 is necessary for this transgenerational effect. All though this experiment is not possible with par-5 RNAi, the authors should either conditionally deplete PAR-5 (such as with auxin inducible degradation) in the P0s, or reword the text and figure to remove these claims.

Minor points:

1.The convention in the field is to only use "transgenerational" to refer to effects that are passed down at least three generations (Perez and Lehner nature cell biology 2019). Effects that are only shown to be passed down a single generation are referred to as "intergenerational". Although the effect shown in this manuscript may indeed be transgenerational, the authors haven't shown this. The authors should either test how many generations this effect lasts, or change the text to clarify that it maybe either intergenerational or transgenerational.

2. Insert in the following sentence "not" after "did": "As shown in Fig 2B, inhibition of aex-5 and eat-2 did elicit pathogen avoidance in glp-1 animals."

3. In Figure 1C and 3A, outlines of the germline are necessary as it is hard to know where the germline is in the current fluorescent images.

4. Other examples of pathogen avoidance being transmitted to progeny have been demonstrated in C. elegans and should be cited:

Moore RS, Kaletsky R & Murphy CT (2019) Piwi/PRG-1 Argonaute and TGF-beta Mediate Transgenerational Learned Pathogenic Avoidance. Cell 177, 1827-1841.e12.

If preprints can be cited:

Kaletsky R, Moore RS, Vrla GD, Parsons LL, Gitai Z & Murphy CT (2020) C. elegans "reads" bacterial non-coding RNAs to learn pathogenic avoidance. bioRxiv, 2020.01.26.920322.

Pereira AG, Gracida X, Kagias K & Zhang Y (2020) C. elegans aversive olfactory learning generates diverse intergenerational effects. bioRxiv, 2020.02.07.939017.

5. There are several instances of "C elegans" which should be "C. elegans".

Rev. 3:

The work of Hong and Aballay aims at contributing to the elucidation of the mechanisms involved in the determination of behavior by signaling from the gut. The particular question the group is addressing is how the intestinal distention caused by bacteria P. aeruginosa in C. elegans mediates avoidance behaviors (by measuring the permanence of animals in the pathogen’s lawn). The authors find that the acetylation of lysine 8 in histone 4 in the gonad is caused by intestinal distention and is essential for both behavioral avoidance and inheritance. Authors highlight a role for the gonad in the intestinal-brain communication axis that triggers behavioral change. This is a fascinating topic of great significance for the field.

I suggest a number of issues need to be addressed:

1. There is no mention in the text of the role of histone modification and specifically of what the H4K8ac is doing transcriptionally. Also, authors do not discuss why this specific modification could be occurring compared to H3K4me3 or H3K4me1, and why were these three selected.

2. Authors used a t-test for their analysis (as mentioned in the methods section). They should instead use a one-way ANOVA with post-hoc analysis for those experiments that contain more than two conditions.

3. Figure 1C. It is really hard to see the gonad in these pictures or distinguish it from any other structure. A bright field or Nomarski picture should be provided and a marker to confirm the localization of the marks. It would be important to show whether neurons are also marked. Can authors explain why animals appear curved?

4. As in point 3, images in Figure S5 need improving. It is hard to distinguish the germline in the photos. Also, there is expression (red color) elsewhere. Which cells are those? It is important that authors show Nomarski images for those staining’s. It will really help to show an independent marker for the gonad or provide clear DAPI images.

5. Figure legends need more detail. For example: Quantification of western blots of extracts from fer-1(b232) animals exposed to E. coli (E. C) or P. aeruginosa (P. A). What is it that is being quantified? Fold change of what? How is this calculated? How many animals in each experiment? In the same line, this should be clarified in the figure axis (applies to all graphs with fold change). For those graphs quantifying intensity of histone acetylation, the same clarification will be important. Are those pixels? Intensity over a control?

6. In the second paragraph of the second section of results reads “As shown in Fig 2B, inhibition of aex-5 and eat-2 did elicit pathogen avoidance in glp-1 animals”. What figure 2B shows is the opposite. I imagine this is a typo (an important one) given what is said afterward.

7. Figure 2C will be very hard to understand for a general audience. I suggest (as mentioned before for the other images) to include a Nomarski image. In these pictures, the glp-1 mutants do not appear equally colonized by PA14-GFP as wild type animals. Additionally, there are other methods that more accurately measure the number of bacteria colonizing the animal intestine (CFU count for instance).

8. The experimental paradigm used in this work does not correspond to transgenerationally inherited phenomena. For an effect to be transgenerational animals should skip at least one generation of encounter to the pathogen and the following generation examined (see Rechavi’s papers for examples of transgenerational paradigms. For transgenerational effects involving pathogens see Palominos et al., 2017 or Moore et al., 2019). The effect observed here could be called intergenerational instead.

9. In the quantifications of immunofluorescence (Supplementary Figures) the n values seem to be random. For example:

S6 Fig n=4

S8 Fig. n=5

S11 Fig n=20

Are these n=4, n=5, n=20 per condition? 4, 5 or 20 on each bacterium or RNAi experiment? Or is it the total of animals screened. In any case, why are the numbers so different?

10. Other points (or simple suggestions):

• I suggest to mention throughout the text the strain of P. aeruginosa used is PA14.

• In the first paragraph of the Results section authors state “P. aeruginosa colonization specifically induces histone H4K8 acetylation in the infected animals”. In the following paragraph it is said that distention alone causes H4K8 acetylation. This apparent contradiction could be avoided by rephrasing the first sentence with “colonization” or other similar word because at that point they do not know whether it is specific to the pathogen.

• It would be nice if all graphs had a homogenous font size.

---

## [Decision Letter · Decision Letter 2]

4 Feb 2021

Dear Dr Aballay,

Thank you for submitting your revised Short Reports entitled "Microbial colonization induces histone acetylation critical for inherited gut-germline-neural signaling" for publication in PLOS Biology. I have now obtained advice from the three original reviewers and have discussed their comments with the Academic Editor. 

Based on the reviews, we will probably accept this manuscript for publication, assuming that you will modify the manuscript to address the data and other policy-related requests noted at the end of this email.

We expect to receive your revised manuscript within two weeks.

-  a cover letter that should detail your responses to any editorial requests.

*Published Peer Review History*

*Early Version*

Sincerely,

Ines

--

Ines Alvarez-Garcia, PhD,

Senior Editor,

PLOS Biology

Fig. 2C, E; Fig. 3B, C, G, H; Fig. 4B; Fig. S1A-D; Fig. S2A-C, E; Fig. S3A-C; Fig. S4A, C; Fig. S5; Fig. S6B and Fig. S7

Reviewers’ comments

Rev. 1:

The authors have thoroughly responded to all reviewers comments. This is a very nice study.

Rev. 2:

The authors have addressed all of my concerns and I now enthusiastically support the publication of the article.

Rev. 3: Andrea Calixto - this reviewer has waived anonymity

The authors have answered all my questions and done the proposed improvements to the manuscript. I especially appreciate all the effort done to perform new experiments which required the inclusion of new authors. This work is an important contribution to the field.

---

## [Editor Report · Decision Letter 3]

27 Feb 2021

Dear Dr Aballay,

Thank you for submitting your revised Short Reports entitled "Microbial colonization induces histone acetylation critical for inherited gut-germline-neural signaling" for publication in PLOS Biology.

We are almost satisfied with the manuscript, but we do have two questions about the data that should be addressed:

Figure S1C: Values in independent assays 1 and 2 are exactly the same. Are these really independent or is it a mistake? If so, please correct it and add the right values.

Figure S4C. There are only 2 replicas – could you please explain why? Please note that we do require 3 replicates for all experiments.

We expect to receive your revised manuscript within one week. 

To submit your revision, please go to https://www.editorialmanager.com/pbiology/ and log in as an Author. Click the link labelled 'Submissions Needing Revision' to find your submission record. Your revised submission must include a cover letter that should detail your responses to any editorial requests.

Sincerely,

Ines

--

Ines Alvarez-Garcia, PhD,

Senior Editor,

PLOS Biology

---

## [Editor Report · Decision Letter 4]

4 Mar 2021

Dear Dr Aballay,

On behalf of my colleagues and the Academic Editor, Jennifer Garrison, I am pleased to say that we can in principle offer to publish your Short Report entitled "Microbial colonization induces histone acetylation critical for inherited gut-germline-neural signaling" in PLOS Biology, provided you address any remaining formatting and reporting issues. These will be detailed in an email that will follow this letter and that you will usually receive within 2-3 business days, during which time no action is required from you. Please note that we will not be able to formally accept your manuscript and schedule it for publication until you have made the required changes.

PRESS

Thank you again for supporting Open Access publishing. We look forward to publishing your paper in PLOS Biology. 

Sincerely, 

Ines

--

Ines Alvarez-Garcia, PhD 

Senior Editor 

PLOS Biology